# Characterization and Biological Activities of Four Biotransformation Products of Diosgenin from *Rhodococcus erythropolis*

**DOI:** 10.3390/molecules28073093

**Published:** 2023-03-30

**Authors:** Yanjie Li, Chengyu Zhang, Kexin Kong, Xiaohui Yan

**Affiliations:** 1State Key Laboratory of Component-Based Chinese Medicine, Tianjin University of Traditional Chinese Medicine, 10 Poyanghu Road, Jinghai District, Tianjin 301617, China; 2Haihe Laboratory of Modern Chinese Medicine, 10 Poyanghu Road, Jinghai District, Tianjin 301617, China

**Keywords:** diosgenin, biotransformation, *Rhodococcus erythropolis*, cyto-protection

## Abstract

Diosgenin (DSG), a steroidal sapogenin derived from the tuberous roots of yam, possesses multiple biological properties. DSG has been widely used as a starting material for the industrial production of steroid drugs. Despite its significant pharmacological activities, moderate potency and low solubility hinder the medicinal application of DSG. Biotransformation is an efficient method to produce valuable derivatives of natural products. In this work, we performed the biotransformation of DSG using five *Rhodococcus* strains. Compounds **1**–**4** were isolated and identified from *Rhodococcus erythropolis*. Compounds **1** and **2** showed potent cytotoxicity against the A549, MCF-7, and HepG2 cell lines. Compounds **3** and **4** are novel entities, and each possesses a terminal carboxyl group attached to the spiroacetal ring. Remarkably, **4** exhibited significant cell protective effects for kidney, liver, and vascular endothelial cells, suggesting the therapeutic potential of this compound in chronic renal diseases, atherosclerosis, and hypertension. We further optimized the fermentation conditions aiming to increase the titer of compound **4**. Finally, the yield of compound **4** was improved by 2.9-fold and reached 32.4 mg/L in the optimized conditions. Our study lays the foundation for further developing compound **4** as a cell protective agent.

## 1. Introduction

Diosgenin (DSG), a natural C_27_ spiroacetal steroid sapogenin, is highly enriched in *Dioscorea zingiberensis*, *Trigonella foenum-graecum,* and *Smilax China* [1]. It is an important raw material for producing more than 200 steroid hormones, such as corticosteroids and sex hormones [2]. DSG also shows potential activities in treating various cancers, cardiovascular diseases, inflammation, and neurological diseases [3,4,5]. However, the clinical application of DSG is impeded by its moderate potency and low aqueous solubility. Numerous DSG derivatives with better pharmacological properties have been synthesized. It was shown that modification of DSG at the C-3 and C-26 positions could modulate its anti-tumor activity [6]. For instance, a DSG derivative with modification at C-3 exhibited better cytotoxicity against HepG2 cells (IC_50_ = 1.9 μM) than DSG (IC_50_ > 10 μM) [7]. The amino acid prodrug strategy can improve the water solubility and oral bioavailability of a compound. The introduction of *_L_*-tryptophan at the C-3 and C-26 of DSG significantly increased its cytotoxicity [8]. The lack of reactive functional groups on the nucleus often restricts the chemical modification of the DSG [9]. With the development of biotechnology, biotransformation has become a powerful tool to generate derivatives of natural compounds. The hydroxylation reactions of DSG by several fungi and bacteria, such as *Syncephalastrum racemosum* [10], *Cunninghamella echinulata* [11], *Bacillus megaterium* [12], *Coriolus versicolor* [13], *Cunninghamella blakesleeana* [14], and *Streptomyces virginiae* IBL-14 [15], occur at the C-9, C-11, C-12, C-15, C-18, and C-25 positions. *Streptomyces virginiae* IBL-14 can also transform DSG to 6-methoxy-6-dehydronuatigenone and 6-dimethoxy-7α-hydroxyldiosgenone [16]. The key intermediates of sex hormones, 4-en-3,17-dione (AD) and androsta-1,4-diene-3,17-dione (androstadienedione, ADD), can be produced in *Rhizopus* spp. by degrading the E and F rings of DSG [17].

*Rhodococcus* is a diverse genus of gram-positive, non-motile, non-sporulating, aerobic bacteria with a high G+C content [18]. *Rhodococcus* can survive in harsh conditions, such as polycyclic aromatic hydrocarbon (PAH)-enriched river sediments [19], groundwater contaminated with nitrophenols and trichloroethene [20], and polychlorinated biphenyl (PCB)-contaminated soil [21]. *Rhodococci* are often used for the bioremediation of contaminants. They are also candidate strains for the biotransformation [22]. For instance, *Rhodococcus* can convert lignin-derived aromatic pollutants into valuable gallate [23] and degrade tetrabromobisphenol A (TBBPA) to a less toxic dimethyl ether diMeO-TBBPA [24]. Through the nitrilase and nitrile hydratase systems, *Rhodococcus* sp. can convert nitriles into carboxylic acid and ammonia [25]. *Rhodococcus* harbor large amounts of enzymes that convert steroids into their derivatives. The cytochrome P450 125 (cyp125) of *Rhodococcus jostii* RHA1 is involved in the formation of steroid C26-carboxylic acid [26]. The cholesterol oxidase (Chox) from *Rhodococcus* sp. NCIM 2891 can biotransform cholesterol into cholestenone [27]. *Rhodococcus erythropolis* can biotransform the inexpensive starting material β-myrcene into a high-value compound geraniol [28]. *Rhodococcus ruber*, *Rhodococcus Globerulus*, and *Rhodococcus coprophilus* can biotransform cortisone and hydrocortisone to their ∆1-dehydrogenated products, prednisone, and prednisolone, which possess better anti-inflammation activity than cortisone and hydrocortisone [29].

In this work, we studied the biotransform of DSG using five *Rhodococcus* species. *R. erythropolis* could convert DSG into four derivatives (compounds **1**–**4**). Compounds **1** and **2** were identified as known compounds diosgenone and 1-dehydrodiosgenone. Compounds **3** and **4** are novel entities, diosgenone-27-oic acid and 1-dehydrodiosgenone-27-oic acid, respectively. These two compounds each possess a terminal carboxyl group attached to the spiroacetal F-ring. We then investigated the biological activities of these four products. Compounds **1** and **2** showed potent cytotoxicity against the A549, MCF-7, and HepG2 cell lines. Compound **4** did not exhibit antitumor activity against the three tumor cell lines. Instead, compound **4** showed cytoprotective activity for various cell lines, which makes it a potential agent to treat chronic renal diseases, atherosclerosis, and hypertension [30]. Based on the time course for the production of these four products, we proposed a pathway for DSG biotransformation in *R. erythropolis*. Ultimately, we improved the titer of compound **4** by 2.9-fold by optimizing the fermentation conditions. Our work lays the foundation for the development of compound **4** as a cell protective agent.

## 2. Results and Discussion

### 2.1. Identification of the Biotransformation Products of DSG

A total of five strains, *Rhodococcus erythropolis*, *Rhodococcus globerulus, Rhodococcus baikonurensis*, *Rhodococcus phenolicus*, and *Rhodococcus aetherivorans,* were used to evaluate their ability to biotransform DSG. Analysis of the fermentation profiles showed that these strains could metabolize DSG (Figure 1A). *R. globerulus* could only convert DSG to produce compound **1** at a low yield. *R. baikonurensis*, *R. phenolicus*, and *R. aetherivorans* all showed the ability to convert DSG to compound **1** and compound **2**. *R. aetherivorans* produced the highest titer of compound **2** compared to the other four strains, which could be used as a producing strain for compound **2**. In particular, *R. erythropolis* showed the best transformation capability among the five strains. It can produce four derivatives of DSG. Therefore, *R. erythropolis* was chosen for large-scale fermentation, and four biotransformation products (compounds **1**–**4**) were isolated and characterized (Figure 1B).

The four biotransformation products of DSG by *R. erythropolis* were purified by using the silica gel column and semi-preparative liquid chromatography. Their structures were identified by Mass spectrometry (MS), 1D and 2D nuclear magnetic resonance (NMR), and Infrared (IR).

Compound **1** was isolated as a white powder. Its molecular formula was determined as C_27_H_40_O_3_ by HR-MS (Appendix A), which gave an ion peak at *m/z* 413.3051 (calcd. for [M + H]^−^: 413.3050). ^1^H NMR and ^13^C NMR data (Appendix A) were compared with the literature to identify compound **1** as diosgenone [31]. Compound **2** was isolated as a white powder. Its molecular formula was determined as C_27_H_38_O_3_ by HR-MS (Appendix A), which gave an ion peak at *m*/*z* 411.2896 (calcd. for [M + H]^−^: 411.2894). Compound **2** was identified as 1-dehydrodiosgenone based on its NMR data (Appendix A) [32]. In previous studies, various microorganisms were found to convert DSG to diosgenone, but only a few strains were able to convert DSG to 1-dehydrodiosgenone [16,31,33]. In *R. erythropolis*, both diosgenone and 1-dehydrodiosgenone are the main biotransformation products.

Compound **3** was isolated as a white powder; [α]^25^_D_ −27.0 (*c* = 0.20, MeOH). The molecular formula of **3** was determined as C_27_H_38_O_5_ by HR-MS analysis (Appendix A), which gave an ion peak at *m*/*z* 443.2789 (calcd. for [M + H]^−^: 443.2792). Different from compound **1** (Appendix A), compound **3** showed C=O and O-H stretching vibration at 1725 cm^−1^ and 3432 cm^−1^, respectively, indicating a carboxyl group in this molecular. We compared the ^1^H NMR and ^13^C NMR spectra of compound **3** (Table 1, Appendix A) with those of compound **1**, which showed that the NMR data of compounds **1** and **3** were highly similar. The ^1^H NMR spectrum of **3** displayed three characteristic methyl signals at *δ* 0.82 (H_3_-18), 1.20 (H_3_-19), and 0.97 (H_3_-21), similar to the signals in compound **1**, but the methyl signal of H_3_-27 disappeared. A comparison of the ^13^C NMR spectrum of **3** with the data of **1** showed the presence of a carboxyl group at *δ* 176.5 (C-27). The ^1^H–^1^H correlation spectroscopy, the heteronuclear multi-bond correlation spectroscopy (HMBC), and heteronuclear single quantum coherence (HSQC) correlations (Figure 2, Appendix A) indicated that the carboxyl group was linked to C-25. The NOESY spectrum (Appendix A) shows that H-25 is an axial bond, so the carboxyl group in compound **3** adopts an equatorial configuration. The structure of **3** was determined to be 25(*R*)-spirosta-4-dien-3-one-27-oic acid and was named diosgenone-27-oic acid. After surveying the public databases, compound **3** was determined as a new compound, which is the carboxylated metabolite of diosgenone at the C27 position.

Compound **4** was isolated as a light-yellow powder, with [α]^25^_D_ −10.0 (*c* = 0.16, MeOH). The molecular formation of compound **4** was determined as C_27_H_36_O_5_ by HR-MS (Appendix A), which gave an ion peak at *m*/*z* 439.2502 (calcd. for [M-H]^−^: 439.2490). The IR spectrum of compound **4** (Appendix A) showed the characteristic absorptions attributable to the ketone carbonyl group (1730 cm^−1^) and the hydroxyl group (3426 cm^−1^) compared with compound **2**. The absorption can be assigned as the characteristic absorption peak of the carboxyl group. The ^1^H NMR and ^13^C NMR spectra of compound **4** (Table 1, Appendix A) are highly similar to those of compound **3**, except for the existence of a double-bond signal [*δ* 7.04 (1H, d, *J* = 10.1 Hz, H-1), 6.23 (1H, dd, *J* = 10.2, 1.0 Hz, H-2)] between C-1 and C-2. Analysis of the ^1^H–^1^H COSY, HMBC, HSQC, and NOESY spectra (Figure 2 and Appendix A) of compound **4** revealed its identity as 25(*R*)-spirosta-1,4-dien-3-one-27-oic acid. Compound **4** was identified as a novel compound by searching the Antibase, NPAtlas, and SciFinder databases, and it was named 1-dehydrodiosgenone-27-oic acid. Single-crystal X-ray crystallography was used to confirm the inferred structure and symmetry space group of the crystals obtained. Purification of **4** from a solution of methanol/dichloromethane (9:1) at room temperature yielded suitable white crystals for X-ray analysis. Compound **4** crystallized in orthorhombic space group P2_1_2_1_2, *a* = 16.3270(9) Å, *b* = 20.3577(11) Å, *c* = 7.0798(4) Å, *V* = 2353.2(2) Å^3^, *Z* = 4, *T* = 170.00 K, μ (GaKα) = 0.430 mm^−1^, *Dcalc* = 1.244 g/cm^3^, 38,910 reflections measured (6.036° ≤ 2Θ ≤ 121.19°), 5390 unique (*R*_int_ = 0.0416, R_sigma_ = 0.0240) which were used in all calculations. The final *R*_1_ was 0.0331 (I > 2σ(I)) and *wR*_2_ was 0.0898. The summarized X-ray crystallographic data of **4** is given in Appendix A, and the crystal structure data has been deposited at the Cambridge Crystallographic Data Centre (CCDC) under deposition number 2245739. The X-ray crystal structure of **4** is shown in Appendix A, confirming the inferred structure of **4**.

### 2.2. Anti-Tumor Activity of the Four DSG Derivatives

Many studies have shown that derivatives of DSG exhibit more potent anti-tumor activity [7]. We investigated the in vitro anti-tumor activities of DSG and the four transformation products obtained in this study, with the anti-cancer drug Doxorubicin HCl as a positive control. Human hepatoma (HepG2), human breast cancer (MCF-7), and human lung carcinoma (A549) cell lines were used for the evaluation. The results are shown in Table 2.

Compared with DSG, compound **1** exhibited cytotoxicity potency against the three cancer cell lines (A549, IC_50_ = 6.26 ± 0.62 μM, MCF-7, IC_50_ = 8.74 ± 0.73 μM, HepG2, IC_50_ = 28.37 ± 2.42 μM). Compound **2** showed significant potency (IC_50_ = 6.16 ± 0.34 μM) against MCF-7 cells, although it was less toxic to A549 (IC_50_ = 13.24 ± 2.13 μM) and HepG2 (IC_50_ = 14.97 ± 0.56 μM) cells. Compound **3** did not exhibit significant anti-tumor activity against A549 cells (IC_50_ = 41.15 ± 2.16 μM) and MCF-7 cells (IC_50_ = 29.68 ± 4.20 μM), and no inhibitory activity against HepG2 cells. Notably, the IC_50_ values of compound **4** were all higher than 100 μM, and the survival rate of **4** was higher than that of normal cultured cells, especially HepG2 cells (Appendix A). Compound **4** increased the cell viability of these three cell lines, indicating that **4** may have a role in promoting cell proliferation. Next, compound **4** was used to investigate its cytoprotective activity.

### 2.3. Cytoprotective Effect of Compound ***4***

There is homeostasis between oxidation and antioxidation, and reactive oxygen species (ROS) accumulation at high levels can cause oxidative damage to biomolecules, leading to cell and organelle damage. Hydrogen peroxide (H_2_O_2_) is the main form of ROS in vivo. H_2_O_2_ is the preferred inducer for cell oxidative stress modeling and was used to induce an oxidative damage model in human embryonic kidney 293T cells [34,35]. Homocysteine (Hcy), a sulfur-containing nonessential amino acid, is a critical intermediate in methionine metabolism [36]. Hcy contains highly reactive sulfur groups, which can rapidly self-oxidize and produce ROS products, such as oxygen free radicals, H_2_O_2,_ and superoxide anions in the presence of oxygen and metal ions [37]. It was significantly correlated with many diseases, such as liver fibrosis, cirrhosis, atherosclerosis, hypertension, and coronary heart disease [38,39]. Previous studies showed that Hcy has a wide range of toxic effects on human hepatocellular carcinoma cells. It is often used to construct the damage model for HepG2 and HUVEC [40]. Therefore, we investigated the cell protective effect of compound **4** by using H_2_O_2_-induced (to 293T) or Hcy-induced (to HepG2 and HUVEC) cell models (Appendix A).

The viability of the 293T cells improved with the increase of compound **4** (Figure 3A). There was a significant difference (*p* < 0.05) in the cell viability between the group treated with 0.1 μM H_2_O_2_ and the model group, indicating that **4** could reduce the damage of 293T cells at 0.1 μM. The cell viability increased to 94.0% in the group with 0.6 μM of **4**. Moreover, there was a significant difference between the group treated with 20 μM and the normal group (*p* < 0.05), and the relative cell viability was increased to 111.4%. As shown in Figure 3B, the viability of HepG2 cells decreased significantly after being treated with Hcy. Adding 0.4 μM of **4** can effectively alleviate the cell damage, and the cell viability was recovered to 93.1% after incubating with 8 μM of **4**. The cell viability was higher than the normal group when 60 μM compound **4** was added to the culture. Similar to the 293T and HepG2 cells, the viability of the HUVEC cells increased apparently by incubating with 0.4 μM of **4** (Figure 3C). Moreover, 6 μM of **4** could restore the cell viability completely. All the above results proved that compound **4** could efficiently protect the kidney, liver, and endothelium from oxidative stress.

Numerous studies have shown that oxidative stress is associated with many diseases, such as cancer, Alzheimer’s disease, diabetes, and arteriosclerosis [30]. In recent years, compounds, such as flavonoids, vitamin E, vitamin C, and resveratrol, have attracted much attention due to their beneficial effects against oxidative stress [41]. Flavonoids in lemon seeds (FLS) were evaluated against H_2_O_2_-induced oxidative damage in 293 T cells. The survival rate of 293T cells increased significantly to 89.2% after treatment with 150 μM of FLS [34]. Resveratrol could alleviate H_2_O_2_-induced HUVEC injury, and 100 μM resveratrol showed the highest cell viability (83.31%) among the range of 20–1000 μM [42]. Both FLS and resveratrol show antioxidation effects at high concentrations. However, compound **4** exerts cell protection against oxidative damage at a low micromolar level without noticeable cytotoxicity until 100 μM. Therefore, these results indicate that **4** may prevent cell apoptosis under oxidative stress, showing the potential for treating diseases caused by oxidative stress damage.

### 2.4. Proposal of Plausible Metabolic Pathways for Compounds ***1***–***4***

Compound **4** is a promising therapeutic agent due to its potent and valuable cytoprotective activity, but the low production titer in fermentation impedes its further development. Therefore, we expected to increase the titer of compound **4**. First, we proposed the plausible metabolic pathways for compounds **1**–**4** in *R. erythropolis* and figured out how compound **4** is produced from DSG. The biotransformation pathways were proposed by analyzing the time course and yields of **1**–**4**. DSG was fermented in *R. erythropolis* from 12 h to 168 h, and the production of the four products was analyzed by UPLC (Appendix A). Compounds **1** and **2** could be readily detected after the addition of DSG into the culture of *R. erythropolis,* and their amounts increased rapidly from 12 h to 72 h. Compound **3** was produced within 24 h, and the content increased significantly from 72 h to 120 h. Compound **4** was detected at 48 h, and its highest titer was achieved at 120 h.

Based on the analysis of fermentation broth at different times, compound **1** (diosgenone) was produced first. The biotransform of DSG to diosgenone involves the C-3 carbonylation and concomitant double-bond migration. This step is the same as the dehydrogenation of cholesterol at the 3β-hydroxyl group by the 3β-hydroxy-steroid dehydrogenase (3β-HSD), which is a prerequisite for cholesterol to be involved in the down-stream degradations [43]. The second DSG derivative accumulated in *R. erythropolis* was compound **2** (1-dehydrodiosgenone), the 1(2)-dehydrogenation product of diosgenone. The 3-ketosteroid Δ1-dehydrogenase (Δ1-KSTD) has been reported to catalyze the 1(2)-dehydrogenation of 3-ketosteroid substrates and plays a crucial role in the production of several steroid drugs, such as 4-androstene-3,17-dione (AD), progesterone, and prednisone [44]. Therefore, 1-dehydrodiosgenone is proposed to be generated from the dehydrogenation of diosgenone catalyzed by KSTD in *Rhodococcus*. Compounds **3** (diosgenone-27-oic acid) and **4** (1-dehydrodiosgenone-27-oic acid) were detected last, and they were supposed to arise from the oxidation of the terminal methyl group at C-27. Several CYP125 enzymes have been characterized to oxidize the terminal methyl groups to carboxyl groups in the cholesterol degradation pathway [26]. Therefore, compounds **3** and **4** are proposed to be biosynthesized by the CYP450 oxidase-catalyzed oxidation of diosgenone and 1-dehydrodiosgenone, respectively. The pathways for the biogenesis of compounds **1**–**4** by *R. erythropolis* were proposed in Figure 4, based on the degradation pathways of cholesterol (Appendix A) [26,45].

### 2.5. Titer Optimization of Compound ***4***

According to the proposed pathway, the conversion of DSG to compound **4** requires the involvement of cytochrome P450 (CYP) oxidases, which are usually expressed at a low level in bacteria. As 5-aminolevulinic acid (5-ALA) and iron ions (ferrous and ferric) are the precursors for heme biosynthesis, their feeding often improves the expression levels of CYP450 enzymes [46]. Therefore, we added ALA and Fe^2+^ (final concentrations 0.5 mg/L) to the fermentation medium, aiming to improve the production of **4**. As expected, the titer of **3** and **4** increased significantly with the addition of ALA and Fe^2+^, while the yields of **1** and **2** decreased (Figure 5A–D). As the first biotransformation product, compound **1** reached its maximum yield at 72 h of fermentation when the yield of the other three compounds was low. With the prolongation of fermentation time, compound **1** was transformed into the other three compounds in the fermentation broth without ALA and Fe^2+^ (Figure 5A). The yield of compound **2** did not change significantly at different fermentation times. After adding the two precursors, the yield of compound **2** decreased significantly without addition, and the yield was highest at 144 h (Figure 5B). The yield of compound **3** increased significantly after adding ALA and Fe^2+^, except for the decrease at 120 h (Figure 5C). Similarly, the titers of the other three compounds were significantly decreased at 120 h compared with the fermentation broth without the addition of ALA and Fe^2+^ (*p*  < 0.001), possibly because compounds **1**, **2**, and **3** were converted to produce more **4** at 120 h. The yield of compound **4** reached the highest at 120 h, whether ALA and Fe^2+^ were added or not. The yield of compound **4** decreased at 144 h and 168 h, indicating that 120 h was the optimal fermentation time to maximize the yield of compound **4** (Figure 5D). Finally, the addition of these two precursors resulted in the titer improvement of compound **4** by 2.9-fold, from 8.25 mg/L to 32.4 mg/L. These results indicated that the enhanced expression of CYP450 enzymes could effectively promote the conversion of **1** and **2** to **3** and **4**.

## 3. Materials and Methods

### 3.1. Instruments

Mass spectrometry (MS) data were collected using liquid chromatography-mass spectrometry ion trap time-of-flight (LCMS-IT-TOF) (Shimadzu, Kyoto, Japan). Nuclear magnetic resonance (NMR) experiments were recorded using a Bruker AV-600 spectrometer (Bruker Biospin, Rheinstetten, Germany). Optical rotations were recorded on a Rudolph Autopol IV (Rudolph Research, Hackettstown, NJ, USA). Infrared (IR) spectra were obtained on Nicolet™ iS™ 10 FTIR spectrometer (Thermo Fisher Scientific, Waltham, MA, USA). Reverse phase Ultra performance liquid chromatography (UPLC) was performed using a Waters Arc2695 UPLC system (Waters, Milford, Fairfield Country, SC, USA) with a Waters CORTECS ^TM^ C18 (4.6 × 50 mm, 2.7 µm) column, equipped with a Waters 2998 UV detector. Column chromatography was performed using silica gel (100–200 mesh, 200–300 mesh, Qingdao Oceanic Chemicals, Qingdao, China). Pre-coated silica gel 60 F_254_ plates (Merck, Darmstadt, Germany) were used for thin-layer chromatography (TCL). Semi-preparative HPLC was carried out on a Waters 1525 pump with a Waters 2998 PDA detector equipped with a Waters XBridge^®^ BEH C18 OBDTM prep column (10 × 150 mm, 5 µm, Waters, Milford, MA, USA).

### 3.2. Chemicals, Strains, and Cells

DSG was purchased from Meryer (Shanghai, China), and a high-purity product was obtained by recrystallization in dichloromethane and methanol. Fetal bovine serum (FBS) was purchased from Thermo Fisher Scientific (Waltham, MA, USA), and the DMEM and RPMI-1640 medium were obtained from Gibco (Thermo Fisher Scientific, Waltham, MA, USA). PS solution (10 KU of Penicillin and 10 mg/L of streptomycin) was purchased from Solarbio (Beijing, China). Analytic solvents were of chromatographic grade, and other reagents and solvents were of analytical grade. *Rhodococcus erythropolis* (CGMCC 1.12549), *Rhodococcus baikonurensis* (CGMCC 1.10292), *Rhodococcus phenolicus* (CGMCC 1.10908), *Rhodococcus aetherivorans* (CGMCC 1.12425), and *Rhodococcus globerulus* (CGMCC 4.1819) were purchased from China General Microbiological Culture Collection Center (CGMCC, Beijing, China). All five strains were cultured in Luria-Bertani (LB) medium containing tryptone (10 g/L), yeast extract (5 g/L), and NaCl (10 g/L). Human cell lines of HepG2 (hepatocarcinoma), MCF-7 (breast cancer), A549 (non-small cell lung cancer), HUVECs (umbilical vein endothelial cells), and 293T (embryonic kidney cells) were obtained from the American Type Culture Collection (ATCC, Manassas, VA, USA). HepG2, MCF-7, and A549 cells were cultured in DMEM medium containing 10% (*v*/*v*) FBS and 1% PS at 37 °C, 5% CO_2_. HUVECs and 293T were cultured in the RPMI-1640 medium with 10% of FBS and 1% PS at 37 °C and 5% CO_2_.

### 3.3. Biotransformation of DSG and Separation of the DSG Derivatives

All five *Rhodococcus* species were cultured using the same method. Preliminary screening of the biotransformation capacity of DSG was carried out in 250 mL flasks containing 50 mL LB liquid medium on a rotary shaker (180 rpm, 28 °C). After inoculation for 24 h, 1.5 mL of DSG solution (pre-dissolved in ethanol, 15 mg/mL) was added into the medium and continued to culture for another 3 days. A scale-up fermentation was performed in 2 L flasks containing 500 mL LB medium with 225 mg DSG. The conditions for control groups were the same as DSG biotransformation groups, except for adding the DSG. The experiment was carried out in triplicate.

After centrifugation of the *Rhodococcus*, the supernatant was separated and extracted twice with an equivalent volume of ethyl acetate. Cells were collected and resuspended via 20 mL of methanol. After being sonicated for 30 min, cells were centrifuged for 10 min at 4000 rpm. The methanol supernatant was dried via an evaporator, then redissolved in an ethyl acetate/water solution (12.5 mL of ethyl acetate and 12.5 mL of water) three times, and all the ethyl acetate layer was collected and combined. Samples were then dried and re-dissolved in methanol and analyzed via a Waters Acr2695 UPLC system (UV 254 nm). The mobile phase consisted of solvent A (0.1% formic acid in deionized water) and solvent B (0.1% formic acid in acetonitrile). The gradient elution conditions were as follows: 5% B from 0 to 3 min, 5–95% B from 3 to 20 min, 95% B from 20 to 24 min, 95–5% B from 24 to 26 min, and 5% B from 26 to 30 min. The flow rate was 0.5 mL/min, and the column temperature was set as 30 °C. The same conditions were used for MS analysis.

After expanding the culture to 500 mL, a total of 3.5 g methanol crude fermentation extract after extraction was evaporated for further purification by silica gel column. TLC and UPLC were used to monitor chromatography fractions (petroleum ether/ethyl acetate mixtures as mobile phase), and detection was under UV at 254 nm. Compound **1** (140 mg) was eluted with a 50:1 mixture of petroleum ether/ethyl acetate, and compound **2** (98 mg) was eluted with a 10:1 mixture of petroleum ether/ethyl acetate. The ethyl acetate/petroleum ether ratio of the mobile phase was continuously increased to pure ethyl acetate to remove impurities in the fermentation extract. Compounds **3** and **4** were collected from the final methanol eluate and then purified via a semi-preparative HPLC, and the separation conditions were as follows: 0.1% formic acid in deionized water was used for mobile phase A, and 0.1% formic acid in acetonitrile was used for mobile phase B. The elution condition was 55% acetonitrile. The flow rate was 3 mL/min, and the injection volume was 300 µL. Finally, compound **3** (7.1mg, t_R_ = 13 min) and compound **4** (3.6mg, t_R_ = 21 min) was purified.

### 3.4. Single-Crystal X-ray Crystallography

Single crystals of compound **4** were obtained by the slow evaporation method from a methanol/dichloromethane (9:1) solution. A suitable crystal was selected and mounted on a diffractometer. The crystal was kept at 170.00 K during data collection. Using Olex2 [47], the structure was solved with the ShelXT [48] structure solution program using Intrinsic Phasing and refined with the ShelXT [49] refinement package using Least Squares minimization.

### 3.5. Cell Viability Assay

The anti-tumor performances of compounds **1**–**4** were evaluated in the HepG2, MCF-7, and A549 cell lines. Cells (3 × 10^4^ cells/mL, 100 µL) were seeded into 96-well plates and incubated for 24 h. Fresh medium containing different concentrations of compounds (0, 0.39, 0.78, 1.56, 3.13, 6.25, 12.5, 25, 50, and 100 µM, respectively) was added into the plates, and cells were incubated for 48 h. Then MTT (5 mg/mL, 20 μL per well) was added into each well, and cells were incubated for another 4 h at 37 °C. After that, the medium was removed, and cells were washed with fresh PBS three times. 100 μL of DMSO was added to the wells, and the absorbance was measured at 492 nm using a microplate reader (Tecan, Swiss, Männedorf, Switzerland), and the IC_50_ values were calculated using GraphPad (Prism 5.0, GraphPad Software, Inc., San Diego, CA, USA).

### 3.6. Cell Protection Assay for Compound ***4***

Firstly, the effect of injurious agents on cell activity was investigated. HepG2, MCF-7, and A549 cells were seeded in 96-well culture plates (3 × 10^4^ cells/mL, 100 µL) and cultured at 37 °C and 5% CO_2_ for 24 h. After that, cells were treated with different concentrations of H_2_O_2_ or Hcy (0–4 mM of H_2_O_2_ to 293T cells and 0–20 mM of Hcy to HepG2, HUVEC cells) for 24 h. The medium was removed, and fresh medium containing 5 mg/mL MTT was readded into the cells, and the cells were washed with PBS three times before adding the DMSO (100 μL per well). The absorbance was measured at 492 nm using a microplate reader, and the cell viability was calculated according to the following formula:$$\mathrm{Cell}\;\mathrm{viability}\;\left(\%\right)=\frac{\left(A2-A0\right)}{\left(A1-A0\right)}\times 100\%$$

A0 refers to the absorbance of blank well, A1 refers to the absorbance of the control group (0 mM of H_2_O_2_ or Hcy), A2 refers to the absorbance of the cells treated with different concentrations of H_2_O_2_ (or Hcy).

After that, the cytoprotecting effects of compound **4** were determined. The HepG2, MCF-7, and A549 cells were seeded in 96-well culture plates (3 × 10^4^ cells/mL, 100 µL) and cultured at 37 °C and 5% CO_2_ for 24 h. Next, the damage regent (0.5 mM of H_2_O_2_ to 293T cells, 12.5 mM of Hcy to HepG2 cells, and 7 mM of Hcy to HUVEC cells) and different concentrations of compound **4** were added into the medium. After 24 h, cells were incubated with fresh medium (containing MTT, 5 mg/mL) for 4 h. Finally, the MTT solution was removed, and cells were washed with PBS three times. 100 μL of DMSO was added into the wells, and the absorbance was measured at 492 nm. The cell viability was calculated according to the above formula.

### 3.7. Statistical Analysis

All tests were repeated in triplicate, and all data were represented as the means ± standard deviations (SD); the data for each group were statistically significant using SPSS version 26.0 (SPSS Inc., Chicago, IL, USA), and a *p*-value of <0.05 was considered statistically significant. Differences between groups were analyzed by one-way analysis of variance (ANOVA) followed by a Tukey test for comparison between groups. Origin 2019b was used for all analyses (Origin Lab Corporation, Northampton, MA, USA).

## 4. Conclusions

In this study, we used a biotransformation strategy to generate DSG derivatives. We performed the biotransformation of DSG using five *Rhodococcus* strains and identified four DSG derivatives from the fermentation broth of *R. erythropolis*. Compounds **3** and **4** are reported for the first time and are formed by the carboxylation of diosgenone and 1-dehydrodiosgenone at C-27. In vitro experiments showed that **1** and **2** have moderate cytotoxicity. Remarkably, compound **4** showed an excellent protective effect on the kidney, liver, and cardiovascular cells in the oxidative damage models, indicating that it is a potential cytoprotective agent. The biotransformation pathways of DSG to compounds **1**–**4** in *R. erythropolis* were proposed by analyzing the time course and yields of **1**–**4**. We further enhanced the expression of P450 enzymes by adding the biosynthetic precursors of heme, aiming to increase the titer of **4**. Finally, the yield of compound **4** was improved by 2.9-fold and reached 32.4 mg/L in the optimized conditions. This study showcases the power of biotransformation in producing bioactive natural product derivatives and lays the foundation for the future development of compound **4** as a cytoprotective agent against oxidative stress.

## Figures and Tables

**Figure 1 molecules-28-03093-f001:**
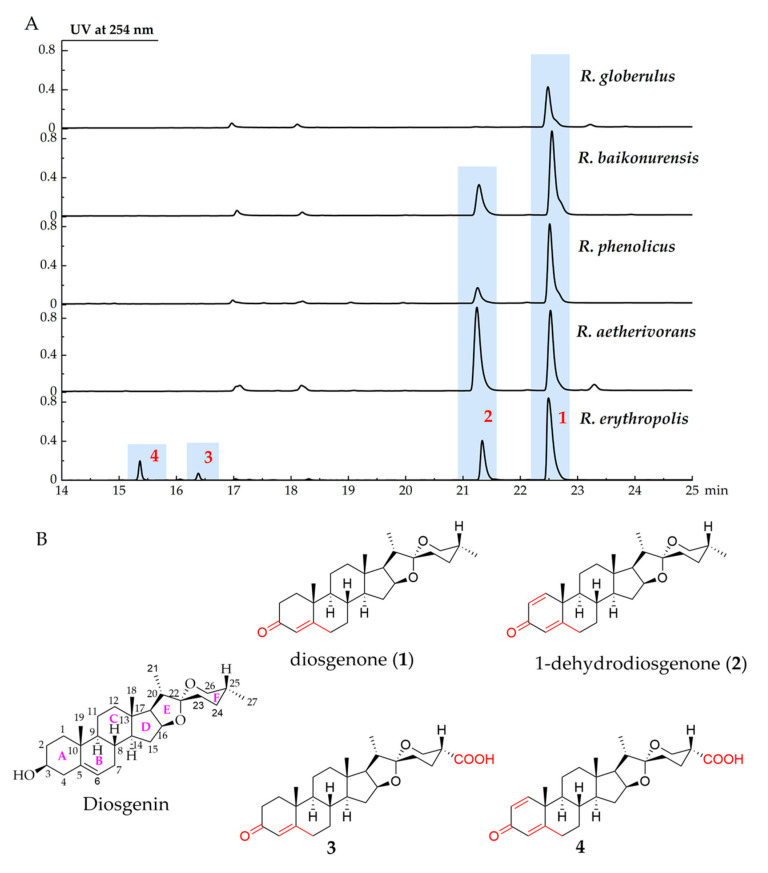
*R. erythropolis* used DSG as a substrate to produce four biotransformation compounds **1**–**4**. (**A**) UPLC profiles of biotransformation of DSG by five *Rhodococcus* strains, (**B**) the structures of DSG and compounds **1**–**4**. The annotations in red in the figure represent the difference between the biotransformation products and DSG.

**Figure 2 molecules-28-03093-f002:**
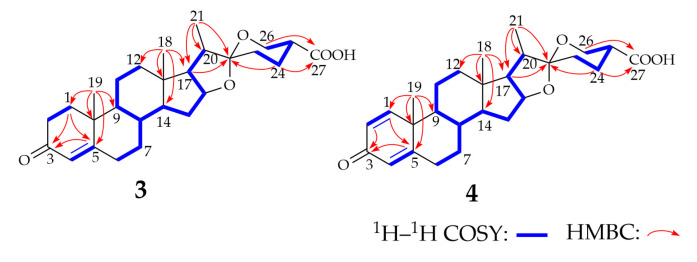
^1^H–^1^H COSY and key HMBC correlations for the structure elucidation of compounds **3** and **4.** The thick blue line in the figure shows the structures observed by the ^1^H–^1^H COSY spectrum. The red arrow shows the structures observed in the HMBC spectrum.

**Figure 3 molecules-28-03093-f003:**
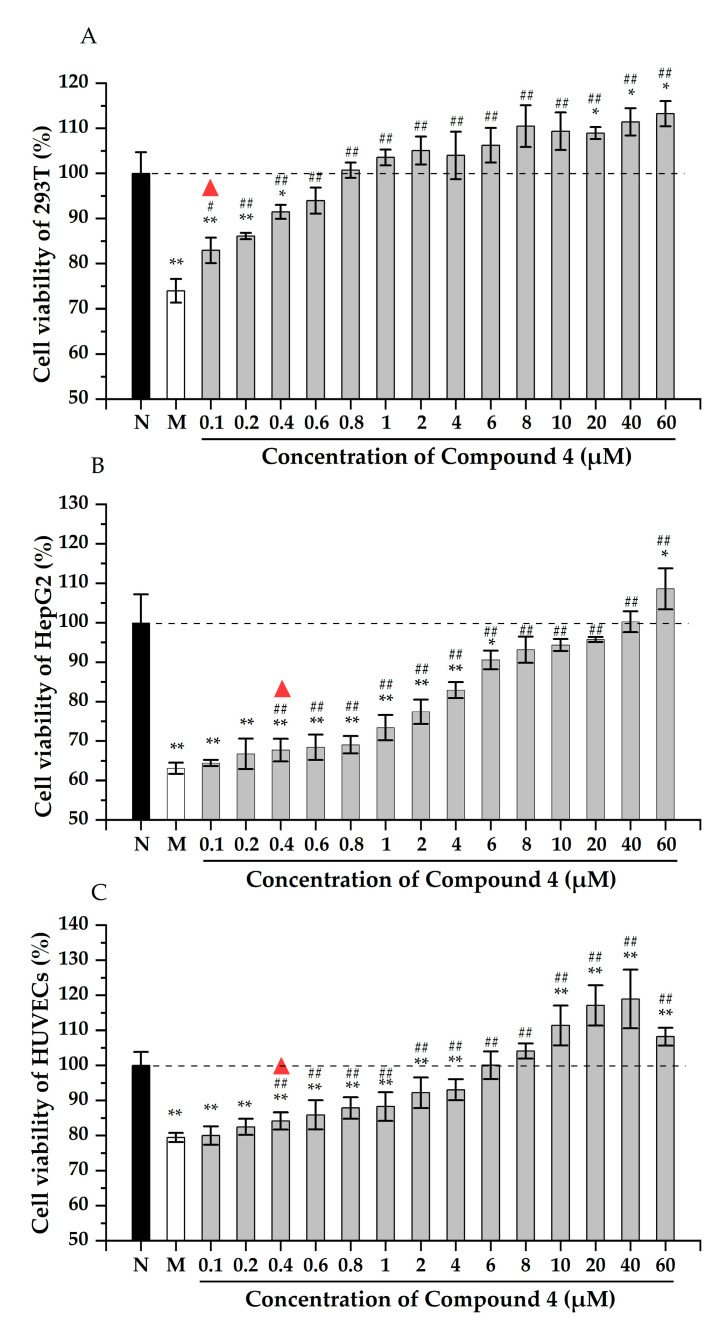
The cytoprotection effect of compound **4** to 293T (**A**), HepG2 (**B**), and HUVECs (**C**) cells. N: normal group, M: model group. Note: * *p* < 0.05, ** *p* < 0.01 vs. normal control; ^#^ *p* < 0.05, ^##^ *p* < 0.01 vs. model control. The triangle indicates that there is a statistical difference between the experimental group and the model group.

**Figure 4 molecules-28-03093-f004:**
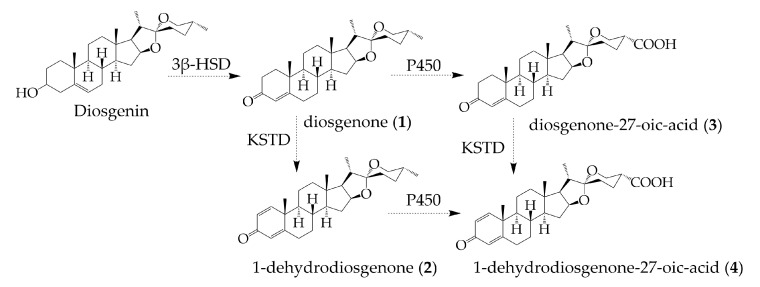
The proposed pathways for the biogenesis of compounds **1**–**4** from DSG in *R. erythropolis*.

**Figure 5 molecules-28-03093-f005:**
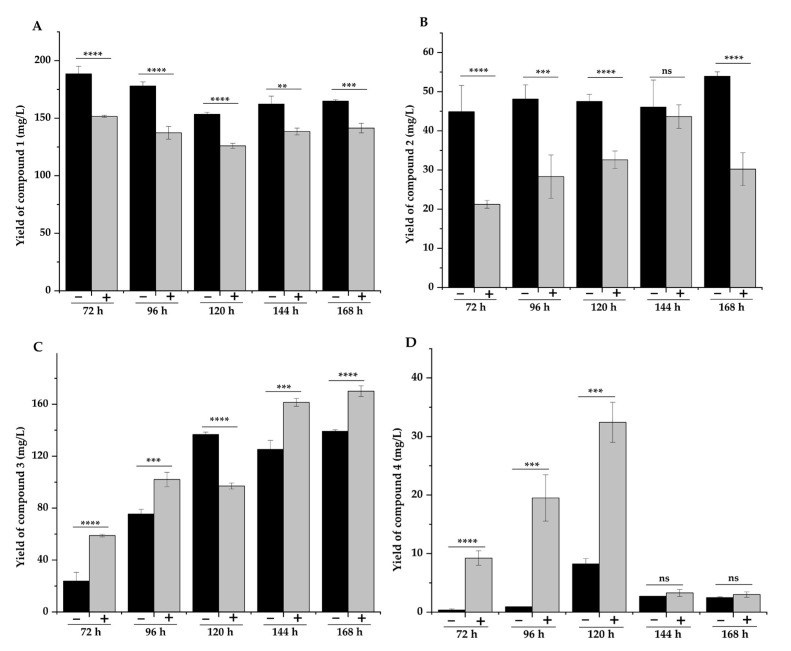
Effect of ALA and Fe^2+^ addition on the yields of compounds **1**–**4.** Compounds **1** (**A**), compounds **2** (**B**), compounds **3** (**C**), and compounds **4** (**D**). Note: “−” represents without the addition of ALA and Fe^2+^, and “+” represents the addition of ALA and Fe^2+^. ** *p*  < 0.01, *** *p*  < 0.005, and **** *p*  < 0.001.

**Table 1 molecules-28-03093-t001:** ^1^H (600 MHz) and ^13^C NMR (150 MHz) spectroscopic data of new compounds **3** and **4** in CDCl_3_.

Position	3		4	
	*δ* _C_	*δ*_H_ (*J* in Hz)	*δ* _C_	*δ*_H_ (*J* in Hz)
1	35.7	2.00 (1H, m); 1.69 (1H, m)	155.9	7.04 (1H, d, *J* = 10.1)
2	34.0	2.35 (1H, m); 2.42 (1H, m)	127.5	6.23 (1H, dd, *J* = 10.2, 1.0)
3	199.6	-	186.4	-
4	123.9	5.73 (1H, s)	123.9	6.08 (1H, s)
5	171.1	-	169.1	-
6	32.8	2.27 (1H, m); 2.41 (1H, m)	32.8	2.38 (1H, m); 2.49 (1H, m)
7	32.1	1.04 (1H, m); 1.86 (1H, m)	31.8	1.34 (1H, m); 2.00 (1H, m)
8	35.2	1.72 (1H, m)	35.2	1.81 (1H, m)
9	53.7	0.94 (1H, m)	52.4	1.05 (1H, m)
10	38.7	-	43.6	-
11	20.8	1.44 (1H, m); 1.53 (1H, m)	22.7	1.67 (1H, m);1.97 (1H, m)
12	39.6	1.16 (1H, m); 1.78 (1H, m)	39.4	1.18 (1H, m); 1.78 (1H, m)
13	40.4	-	40.7	-
14	55.7	1.12 (1H, m)	55.3	1.11 (1H, m)
15	31.6	1.32 (1H, m); 2.00 (1H, m)	33.7	1.07 (1H, m); 1.96 (1H, m)
16	80.9	4.40 (1H, dd, *J* = 14.9, 7.6)	80.8	4.40 (1H, dd, *J* = 14.8, 7.5)
17	62.0	1.76 (1H, m)	61.9	1.77 (1H, m)
18	16.3	0.82 (3H, s)	16.4	0.85 (3H, s)
19	17.4	1.20 (3H, s)	18.8	1.25 (3H, s)
20	41.6	1.91 (1H, t, *J* = 6.9)	41.6	1.91 (1H, t, *J* = 7.0)
21	14.4	0.97 (3H, d, *J* = 6.9)	14.4	0.97 (3H, d, *J* = 6.9)
22	108.8	-	108.8	-
23	30.2	1.69 (2H, m)	30.2	1.69 (2H, m)
24	23.0	2.00 (2H, m)	23.0	2.00 (2H, m)
25	40.5	2.60 (1H, m)	40.5	2.60 (1H, m)
26	60.6	3.83 (1H, dd, *J* = 22.2, 11.0); 3.82 (1H, dd, *J* = 18.1, 11.2)	60.7	3.82 (1H, dd, J = 21.4, 10.9); 3.81 (1H, dd, J = 14.9, 10.6)
27	176.5	-	176.2	-

*δ* in ppm, *J* in Hz. S—singlet; d—doublet; t—triplet; m—multiple.

**Table 2 molecules-28-03093-t002:** In vitro cytotoxicity of DSG and compounds **1–4**.

		IC_50_ (μM)	
	A549	MCF-7	HepG2
Diosgenin	>100	45.49 ± 2.58	59.02 ± 2.88
**1**	6.26 ± 0.62	8.74 ± 0.73	28.37 ± 2.42
**2**	13.24 ± 2.13	6.16 ± 0.34	14.97 ± 0.56
**3**	41.15 ± 2.16	29.68 ± 4.20	>100
**4**	>100	>100	>100
Doxorubicin HCl	4.00 ± 0.07	4.68 ± 0.21	1.39 ± 0.16

IC_50_ represents the concentration of the compound measured using MTT assay when 50% of the cells were inhibited. Data represent mean values ± standard deviations for three independent experiments.

## Data Availability

The data presented are available in the manuscript and Appendix A.

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
