# Peer review of "Characterization and Biological Activities of Four Biotransformation Products of Diosgenin from Rhodococcus erythropolis"

_molecules, 2023, doi:10.3390/molecules28073093_

Round 1

Reviewer 1 Report

1- I advise the author to perform an X-ray analysis of one of compound 3 or 4

2- No in vitro study in the manuscript to detect the molecular target of these compounds as anti-cancer (SKp2 inhibitor for example) or even Apoptosis

3- in Table 2 (in vitro cytotoxicity), the author should use another reference compound other than Diosgenin 

Author Response

Dear reviewer,

We are pleased to submit our revised manuscript entitled "Characterization and Biological Activities of Four Biotransformation Products of Diosgenin from Rhodococcus erythropolis" (manuscript ID: molecules-2191262) for your reconsideration for publication in Molecules.

We would like to express our gratitude for the thorough review and valuable comments provided by you. The feedback has been instrumental in enhancing the quality of our article. We apologize for the delay in responding to your comments. It took longer than expected to obtain the required experimental data, which caused the delay. We hope to have your understanding. We have addressed your comments by incorporating new data, which we have detailed below, and revising the manuscript accordingly. We have used "Track Changes" to highlight the revisions made in the paper, and we have attached detailed responses to each comment. As we can only attach one document to this reply, we have attached our revised manuscript. We have also sent the revised manuscript and revised supporting information to the editor by mail.

Thank you for your time and consideration. We look forward to hearing the final decision on the revised manuscript.

Yours sincerely,

Xiaohui Yan

Response table to Reviewer #1

No.

Reviewer comment

Author response

Change made

1

1. I advise the author to perform an X-ray analysis of one of compound 3 or 4

Thanks for the reviewer’s suggestion. As reviewer pointed out, we have performed an X-ray analysis of compound 4. We added relevant data information in the manuscript and in the supporting information. The changes in the text were labeled via "Track Changes" format.

Page 4, line 151–154;

Page 12, line 366–372

Supporting information, Table S2

2

2. No in vitro study in the manuscript to detect the molecular target of these compounds as anti-cancer (SKp2 inhibitor for example) or even Apoptosis

We sincerely thank the reviewer for careful reading. Many studies have shown that DSG derivatives have with better pharmacological properties, and their signaling pathways have been studied. We presented some studies on the molecular target of DSG derivatives as anti-cancer (SKp2 inhibitor for example) as background in the original manuscript. Based on your comments, we realize that this may be misleading to our readers. Therefore, we have deleted and rewritten this part in the revised manuscript, and we are deeply sorry for the misunderstanding. This part is used to show that these compounds have anti-cancer activity, and mainly showed the cytoprotective activity of compound 4 and subsequent research of compound 4. Therefore, the contents of this chapter can support the experimental conclusions in this article. On the basis of showing that these compounds have anti-cancer effects, we will study the molecular targets of these compounds in the future experiments. The changes in the text were labeled via "Track Changes" format.

Page 6, line 164–169

3

3. in Table 2 (in vitro cytotoxicity), the author should use another reference compound other than Diosgenin

Thanks for the reviewer’s suggestion. As reviewer pointed out, we chose the anticancer drug Doxorubicin HCl as our positive drug. The changes in the text were labeled via "Track Changes" format.

Page 6, line 170-180

Supporting information, Figure S24

Reviewer 2 Report

This manuscript describes the biotransformation of DSG into two reported (diosgenone & 1-dehydrodiosgenone) and two new (diosgenone-27-oic acid & 1-dehydrodiosgenone-oic acid) compounds using R. erythropolis bacterial strain. Whereas compounds diosgenone & 1-dehydrodiosgenone showed potent cytotoxic activity, 1-dehydrodiosgenone-oic acid showed cytoprotective activity. My major concerns are as follows:

 1.      Provide the details of the silica gel column and semi-preparative liquid chromatography.

2.      Fig. S8/S16: Provide full scale (4000-400 cm-1) IR spectrum.

3.      I suggest including overlaid IR spectra of compounds 1 & 3 and 2 & 4. Accordingly, the discussion should be modified.

4.      Authors are suggested to include integrated and labelled NMR spectra of the compounds. Peaks which are not relevant to the structure should be identified and labelled.

5.      There is almost no discussion on the bioactivity of compound 3. It should be included in the revised MS.

Author Response

Dear reviewer,

Please find enclosed our revised manuscript entitled “Characterization and Biological Activities of Four Biotransformation Products of Diosgenin from Rhodococcus erythropolis” (manuscript ID:molecules-2191262), which is revised for consideration of publication in Molecules.

We would like to thank the reviewers for thoroughly reviewing our manuscript and making many thoughtful comments. These comments are valuable and helpful for improving our article. We have added some new data, described in detail below, and revised the manuscript to address reviewers’ comments. We have made revision using "Track Changes" in this paper and the detailed responses to each comment were attached. Since only one document can be attached to this reply, we have attached our revised manuscript. We sent the revised manuscript and additional documents to the editor by mail.

I am looking forward to hearing the final decision regarding on the revised manuscript.

Yours sincerely,

Xiaohui Yan

No.

Reviewer comment

Author response

 Change made

1

1. Provide the details of the silica gel column and semi-preparative liquid chromatography.

Thanks for the reviewer’s suggestion. As reviewer pointed out, we have added more details of the silica gel column and semi-preparative liquid chromatography. The changes in the text were labeled via "Track Changes" format.

Page 10, line 311-312;

Page 11, line 354-357, line 359-361, line 364-365, line366

2

2. Fig. S8/S16: Provide full scale (4000-400 cm-1) IR spectrum.

Thanks for the reviewer’s suggestion. As suggested by the reviewer, we redetermined the IR spectrum of compounds 3 and 4, and provided full scale (4000-400 cm-1) IR spectrum in Fig. S8/S16. The changes in the text were labeled via "Track Changes" format.

Page 10, line 305-306, line 311-312;

Figure S8, Figure S16

3

3. I suggest including overlaid IR spectra of compounds 1 & 3 and 2 & 4. Accordingly, the discussion should be modified.

Thanks for the reviewer’s suggestion. As reviewer pointed out, we determined the IR spectra of compounds 1 and 3, and compared the IR spectra of compounds 1 & 3 and 2 & 4. The discussion of the relative infrared in the manuscript has been modified according to the spectrum of the comparison. The changes in the text were labeled via "Track Changes" format.

Page 3, line 119-122; Page 4, line 138-142;

Figure S8, Figure S16

4

4. Authors are suggested to include integrated and labelled NMR spectra of the compounds. Peaks which are not relevant to the structure should be identified and labelled.

We sincerely thank the reviewer for careful reading. As suggested by the reviewer, we have integrated and labelled NMR spectra of the four compounds and unrelated peaks to the structures. We have revised the data of compounds 14 and the relevant contents in the manuscript. The changes in the text were labeled via "Track Changes" format.

Figure S2, Figure S5, Figure S9, Figure S17, Table S1, Table 1

Page 4, line 143-145

5

5. There is almost no discussion on the bioactivity of compound 3. It should be included in the revised MS

We sincerely appreciate the valuable comments and we have added the bioactivity of compound 3 to the manuscript. The changes in the text were labeled via "Track Changes" format.

Page 6, line 171-177

Round 2

Reviewer 2 Report

The authors have revised their manuscript significantly but not satisfactorily. I have noted that the revision has not been done entirely as per the comments raised by the reviewer. Some issues which require further attention are:

 1.     Full scale (4000-400 cm-1) IR spectrum has not been provided as suggested last time. Therefore, the presence of the carboxyl group is not evident from the supplied data. Authors must provide it.

 2.     There are no overlaid IR spectra found in the MS or SI.

 3.     NMR spectra are not labelled and integrated.

 4.     I also suggest that the authors expand the discussion on X-ray structure.

Thank you very much for the revised files. I have checked the files and found that revision has been done as per the suggestion. Therefore the paper can be accepted now.